# Keeping Your Distance: Solving Sparse Reward Tasks Using Self-Balancing Shaped Rewards

**Alexander Trott**
Salesforce Research
atrott@salesforce.com

**Stephan Zheng**
Salesforce Research
stephan.zheng@salesforce.com

**Caiming Xiong**
Salesforce Research
cxiong@salesforce.com

**Richard Socher**
Salesforce Research
rsocher@salesforce.com

## Abstract

While using shaped rewards can be beneficial when solving sparse reward tasks, their successful application often requires careful engineering and is problem specific. For instance, in tasks where the agent must achieve some goal state, simple distance-to-goal reward shaping often fails, as it renders learning vulnerable to local optima. We introduce a simple and effective model-free method to learn from shaped distance-to-goal rewards on tasks where success depends on reaching a goal state. Our method introduces an auxiliary distance-based reward based on *pairs* of rollouts to encourage diverse exploration. This approach effectively prevents learning dynamics from stabilizing around local optima induced by the naive distance-to-goal reward shaping and enables policies to efficiently solve sparse reward tasks. Our augmented objective does not require any additional reward engineering or domain expertise to implement and converges to the original sparse objective as the agent learns to solve the task. We demonstrate that our method successfully solves a variety of hard-exploration tasks (including maze navigation and 3D construction in a Minecraft environment), where naive distance-based reward shaping otherwise fails, and intrinsic curiosity and reward relabeling strategies exhibit poor performance.

## 1 Introduction

Reinforcement Learning (RL) offers a powerful framework for teaching an agent to perform tasks using only observations from its environment. Formally, the goal of RL is to learn a policy that will maximize the reward received by the agent; for many real-world problems, this requires access to or engineering a reward function that aligns with the task at hand. Designing a well-suited *sparse* reward function simply requires defining the criteria for solving the task: reward is provided if the criteria for completion are met and withheld otherwise. While designing a suitable sparse reward may be straightforward, learning from it within a practical amount of time often is not, often requiring exploration heuristics to help an agent discover the sparse reward (Pathak et al., 2017; Burda et al., 2018b,a). *Reward shaping* (Mataric, 1994; Ng et al., 1999) is a technique to modify the reward signal, and, for instance, can be used to relabel and learn from failed rollouts, based on which ones made more progress towards task completion. This may simplify some aspects of learning, but whether the learned behavior improves task performance depends critically on careful design of the shaped reward (Clark & Amodei, 2016). As such, reward shaping requires domain-expertise and is often problem-specific (Mataric, 1994).

Tasks with well-defined goals provide an interesting extension of the traditional RL framework (Kaelbling, 1993; Sutton et al., 2011; Schaul et al., 2015). Such tasks often require RL agents to deal with goals that vary across episodes and define success as achieving a state within some distance of the episode's goal. Such a setting naturally defines a sparse reward that the agent receives when it achieves the goal. Intuitively, the same distance-to-goal measurement can be further used for reward shaping (without requiring additional domain-expertise), given that it measures progress towards success during an episode. However, reward shaping often introduces new local optima that can prevent agents from learning the optimal behavior for the original task. In particular, the existence and distribution of local optima strongly depends on the environment and task definition.

As such, successfully implementing reward shaping quickly becomes problem specific. These limitations have motivated the recent development of methods to enable learning from sparse rewards (Schulman et al., 2017; Liu et al., 2019), methods to learn latent representations that facilitate shaped reward (Ghosh et al., 2018; Nair et al., 2018; Warde-Farley et al., 2019), and learning objectives that encourage diverse behaviors (Haarnoja et al., 2017; Eysenbach et al., 2019).

We propose a simple and effective method to address the limitations of using distance-to-goal as a shaped reward. In particular, we extend the naive distance-based shaped reward to handle *sibling* trajectories, pairs of independently sampled trajectories using the same policy, starting state, and goal. Our approach, which is simple to implement, can be interpreted as a type of self-balancing reward: we encourage behaviors that make progress towards the goal and simultaneously use sibling rollouts to estimate the local optima and encourage behaviors that avoid these regions, effectively balancing exploration and exploitation. This objective helps to *de*-stabilize local optima without introducing new stable optima, preserving the task definition given by the sparse reward. This additional objective also relates to the entropy of the distribution of terminal states induced by the policy; however, unlike other methods to encourage exploration (Haarnoja et al., 2017), our method is "self-scheduling" such that our proposed shaped reward converges to the sparse reward as the policy learns to reach the goal.

Our method combines the learnability of shaped rewards with the generality of sparse rewards, which we demonstrate through its successful application on a variety of environments that support goal-oriented tasks. In summary, our contributions are as follows:

- We propose Sibling Rivalry, a method for model-free, dynamic reward shaping that preserves optimal policies on sparse-reward tasks.
- We empirically show that Sibling Rivalry enables RL agents to solve hard-exploration sparse-reward tasks, where baselines often struggle to learn. We validate in four settings, including continuous navigation and discrete bit flipping tasks as well as hierarchical control for 3D navigation and 3D construction in a demanding Minecraft environment.

## 2  Preliminaries

Consider an agent that must learn to maximize some task reward through its interactions with its environment. At each time point $t$ throughout an episode, the agent observes its state $s_t \in S$ and selects an action $a_t \in A$ based on its policy $\pi(a_t|s_t)$, yielding a new state $s_t'$ sampled according to the environment's transition dynamics $p(s_t'|s_t, a_t)$ and an associated reward $r_t$ governed by the task-specific reward function $r(s_t, a_t, s_t')$. Let $\boldsymbol{\tau} = \{(s_t, a_t, s_t', r_t)\}_{t=0}^{T-1}$ denote the trajectory of states, actions, next states, and rewards collected during an episode of length $T$, where $T$ is determined by either the maximum episode length or some task-specific termination conditions. The objective of the agent is to learn a policy that maximizes its expected cumulative reward: $\mathbb{E}_{\boldsymbol{\tau} \sim \pi, p} [\Sigma_t \gamma^t r_t]$.

**Reinforcement Learning for Goal-oriented tasks.**  The basic RL framework can be extended to a more general setting where the underlying association between states, actions, and reward can change depending on the parameters of a given episode (Sutton et al., 2011). From this perspective, the agent must learn to optimize a *set* of potential rewards, exploiting the shared structure of the individual tasks they each represent. This is applicable to the case of learning a *goal-conditioned* policy $\pi(a_t|s_t, g)$. Such a policy must embed a sufficiently generic understanding of its environment to choose whatever actions lead to a state consistent with the goal $g$ (Schaul et al., 2015). This setting naturally occurs whenever a task is defined by some set of goals $G$ that an agent must learn to reach when instructed. Typically, each episode is structured around a specific goal $g \in G$ sampled from the

task distribution. In this work, we make the following assumptions in our definition of "goal-oriented task":

1. The task defines a distribution over starting states and goals $\rho(s_0, g)$ that are sampled to start each episode.

2. Goals can be expressed in terms of states such that there exists a function $m(s) : S \to G$ that maps state $s$ to its equivalent goal.

3. $S \times G \to \mathbb{R}^+$ An episode is considered a success once the state is within some radius of the goal, such that $d(s, g) \leq \delta$, where $d(x, y) : G \times G \to \mathbb{R}^+$ is a distance function[1] and $\delta \in \mathbb{R}^+$ is the distance threshold. (Note: this definition is meant to imply that the distance function internally applies the mapping $m$ to any states that are used as input; we omit this from the notation for brevity.)

This generic task definition allows for an equally generic sparse reward function $r(s, g)$:

$$r(s, g) = \begin{cases} 1, & d(s, g) \leq \delta \\ 0, & \text{otherwise} \end{cases} \tag{1}$$

From this, we define $r_t \triangleq r(s'_t, g)$ so that reward at time $t$ depends on the state reached after taking action $a_t$ from state $s_t$. Let us assume for simplicity that an episode terminates when either the goal is reached or a maximum number of actions are taken. This allows us to define a single reward for an entire trajectory considering only the terminal state, giving: $r_{\boldsymbol{\tau}} \triangleq r(s_T, g)$, where $s_T$ is the state of the environment when the episode terminates. The learning objective now becomes finding a goal-conditioned policy that maximizes $\mathbb{E}_{\boldsymbol{\tau} \sim \pi, p, \, s_0, g \sim \rho} [r_{\boldsymbol{\tau}}]$.

## 3  Approach

**Distance-based shaped rewards and local optima.**   We begin with the observation that the distance function $d$ (used to define goal completion and compute sparse reward) may be exposed as a shaped reward without any additional domain knowledge:

$$\tilde{r}(s, g) = \begin{cases} 1, & d(s, g) \leq \delta \\ -d(s, g), & \text{otherwise} \end{cases}, \qquad \tilde{r}_{\boldsymbol{\tau}} \triangleq \tilde{r}(s_T, g). \tag{2}$$

By definition, a state that globally optimizes $\tilde{r}$ also achieves the goal (and yields sparse reward), meaning that $\tilde{r}$ preserves the global optimum of $r$. While we expect the distance function itself to have a single (global) optimum with respect to $s$ and a fixed $g$, in practice we need to consider the possibility that other *local* optima exist because of the state space structure, transition dynamics and other features of the environment. For example, the agent may need to *increase* its distance to the goal in order to eventually reach it. This is exactly the condition faced in the toy task depicted in Figure 1. We would like to gain some intuition for how the learning dynamics are influenced by such local optima and how this influence can be mitigated.

The "learning dynamics" refer to the interaction between (i) the distribution of terminal states $\rho_g^\pi(s_T)$ induced by a policy $\pi$ in pursuit of goal $g$ and (ii) the optimization of the policy with respect to $\mathbb{E}_{\rho_g^\pi(s_T)}[\tilde{r}(s_T, g)]$. A local optimum $o_g \in S$ can be considered "stable" if, for all policies within some basin of attraction, continued optimization causes $\rho_g^\pi(s_T)$ to converge to $o_g$. Figure 1 (middle) presents an example of this. The agent observes its 2D position along the track and takes an action to change its position; its reward is based on its terminal state (after 5 steps). Because of its starting position, maximizing the naive reward $\tilde{r}(s, g)$ causes the policy to "get stuck" at the local optimum $o_g$, i.e., the final state $\rho_g^\pi(s_T)$ is peaked around $o_g$.

In this example, the location of the local optimum is obvious and we can easily engineer a reward bonus for avoiding it. In its more general form, this augmented reward is:

$$r'(s, g, \bar{g}) = \begin{cases} 1, & d(s, g) \leq \delta \\ \min\left[0, -d(s, g) + d(s, \bar{g})\right], & \text{otherwise} \end{cases}, \qquad r'_{\boldsymbol{\tau}} \triangleq r'(s_T, g, \bar{g}). \tag{3}$$

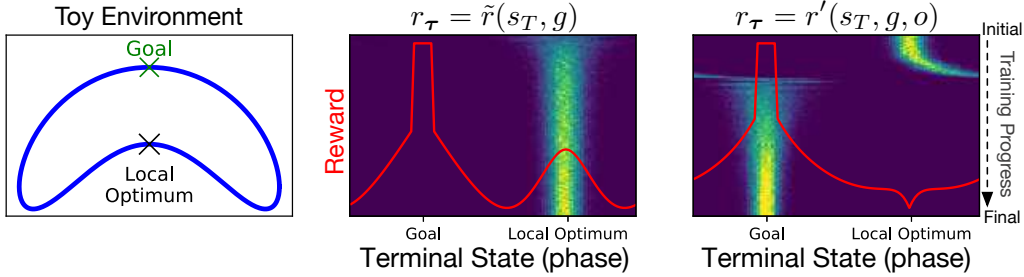

Figure 1: **Motivating example.** (Left) The agent's task is to reach the goal (green X) by controlling its position along a warped circular track. A distance-to-goal reward ($L_2$ distance) creates a local optimum $o_g$ (black X). (Middle and Right) Terminal state distributions during learning. The middle figure shows optimization using a distance-to-goal shaped reward. For the right figure, the shaped reward is augmented to include a hand-engineered bonus for avoiding $o_g$ (Eq. 3; $\bar{g} \leftarrow o_g$). The red overlay illustrates the reward at each phase of the track.

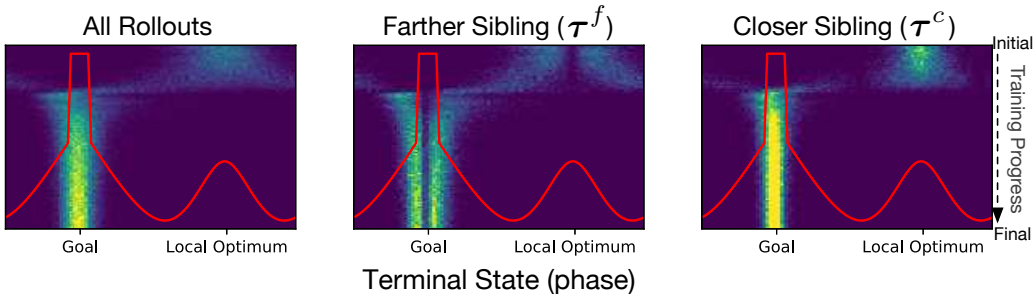

Figure 2: **Learning with Sibling Rivalry.** Terminal state distribution over training when using SR. Middle and right plots show the farther $\tau^f$ and closer $\tau^c$ trajectories, respectively. Red overlay illustrates the shape of the naive distance-to-goal reward $\tilde{r}$.

where $\bar{g} \in G$ acts as an 'anti-goal' and specifies a state that the agent should avoid, e.g., the local optimum $o_g$ in the case of the toy task in Figure 1. Indeed, using $r'$ and setting $\bar{g} \leftarrow o_g$ (that is, using $o_g$ as the 'anti-goal'), prevents the policy from getting stuck at the local optimum and enables the agent to quickly learn to reach the goal location (Figure 1, right).

While this works well in this toy setting, the intuition for which state(s) should be used as the 'anti-goal' $\bar{g}$ will vary depending on the environment, the goal $g$, and learning algorithm. In addition, using a fixed $\bar{g}$ may be self-defeating if the resulting shaped reward introduces its own new local optima. To make use of $r'(s, g, \bar{g})$ in practice, we require a method to dynamically estimate the local optima that frustrate learning without relying on domain-expertise or hand-picked estimations.

**Self-balancing reward.**   We propose to estimate local optima directly from the behavior of the policy by using *sibling* rollouts. We define a pair of sibling rollouts as two independently sampled trajectories sharing the same starting state $s_0$ and goal $g$. We use the notation $\tau^f, \tau^c \sim \pi | g$ to denote a pair of trajectories from 2 sibling rollouts, where the superscript specifies that $\tau^c$ ended closer to the goal than $\tau^f$, i.e. that $\tilde{r}_{\tau^c} \geq \tilde{r}_{\tau^f}$. By definition, optimization should tend to bring $\tau^f$ closer towards $\tau^c$ during learning. That is, it should make $\tau^f$ less likely and $\tau^c$ more likely. In other words, the terminal state of the closer rollout $s_T^c$ can be used to estimate the location of local optima created by the distance-to-goal shaped reward.

To demonstrate this, we revisit the toy example presented in Figure 1 but introduce paired sampling to produce sibling rollouts (Figure 2). As before, we optimize the policy using $r'$ but with 2 important modifications. First, we use the sibling rollouts for *mutual relabeling* using the augmented shaped

---
**Algorithm 1:** Sibling Rivalry
---
**Given**

- Environment, Goal-reaching task w/ $S, G, A, \rho(s_0, g), m(), d(,), \delta$ and max episode length
- Policy $\pi : S \times G \times A \to [0, 1]$ and Critic $V : S \times G \times G \to \mathbb{R}$ with parameters $\theta$
- On-policy learning algorithm $\mathbb{A}$, e.g., REINFORCE, Actor-critic, PPO
- Inclusion threshold $\epsilon$

**for** *iteration = 1...K* **do**
    Initialize transition buffer $D$
    **for** *episode = 1...M* **do**
        Sample $s_0, g \sim \rho$
        $\boldsymbol{\tau}^a \leftarrow \pi_\theta(...)|_{s_0,g}$    # Collect rollout
        $\boldsymbol{\tau}^b \leftarrow \pi_\theta(...)|_{s_0,g}$    # Collect sibling rollout
        Relabel $\boldsymbol{\tau}^a$ reward using $r'$ and $\bar{g} \leftarrow m(s_T^b)$
        Relabel $\boldsymbol{\tau}^b$ reward using $r'$ and $\bar{g} \leftarrow m(s_T^a)$
        **if** $d(s_T^a, g) < d(s_T^b, g)$ **then**
            $\boldsymbol{\tau}^c \leftarrow \boldsymbol{\tau}^a$
            $\boldsymbol{\tau}^f \leftarrow \boldsymbol{\tau}^b$
        **else**
            $\boldsymbol{\tau}^c \leftarrow \boldsymbol{\tau}^b$
            $\boldsymbol{\tau}^f \leftarrow \boldsymbol{\tau}^a$
        **if** $d(s_T^c, s_T^f) < \epsilon$ **or** $d(s_T^c, g) < \delta$ **then**
            Add $\boldsymbol{\tau}^f$ and $\boldsymbol{\tau}^c$ to buffer $D$
        **else**
            Add $\boldsymbol{\tau}^f$ to buffer $D$
    Apply on-policy algorithm $\mathbb{A}$ to update $\theta$ using examples in $D$
---

reward $r'$ (Eq. 3), where each rollout treats its sibling's terminal state as its own anti-goal:

$$r'_{\boldsymbol{\tau}^f} = r'(s_T^f, g, s_T^c) \quad \& \quad r'_{\boldsymbol{\tau}^c} = r'(s_T^c, g, s_T^f). \tag{4}$$

Second, we only include the closer-to-goal trajectory $\boldsymbol{\tau}^c$ for computing policy updates if it reached the goal. As shown in the distribution of $s_T^c$ over training (Figure 2, right), $s_T^c$ remains closely concentrated around *an* optimum: the local optimum early in training and later the global optimum $g$. Our use of sibling rollouts creates a reward signal that intrinsically balances exploitation and exploration by encouraging the policy to minimize distance-to-goal while de-stabilizing local optima created by that objective. Importantly, as the policy converges towards the *global* optimum (i.e. learns to reach the goal), $r'$ converges to the original underlying sparse reward $r$.

**Sibling Rivalry.** From this, we derive a more general method for learning from sibling rollouts: Sibling Rivalry (SR). Algorithm 1 describes the procedure for integrating SR into existing on-policy algorithms for learning in the settings we described above[2]. SR has several key features:

1. sampling sibling rollouts,
2. mutual reward relabeling based on our self-balancing reward $r'$,
3. selective exclusion of $\boldsymbol{\tau}^c$ (the closer rollout) trajectories from gradient estimation, using hyperparameter $\epsilon \in \mathbb{R}^+$ for controlling the inclusion/exclusion criterion.

Consistent with the intuition presented above, we find that ignoring $\boldsymbol{\tau}^c$ during gradient estimation helps prevent the policy from converging to local optima. In practice, however, it can be beneficial to learn directly from $\boldsymbol{\tau}^c$. The hyperparameter $\epsilon$ serves as an inclusion threshold for controlling when $\boldsymbol{\tau}^c$ is included in gradient estimation, such that SR always uses $\boldsymbol{\tau}^f$ for gradient estimation and includes $\boldsymbol{\tau}^c$ only if it reaches the goal or if $d(s_T^f, s_T^c) \leq \epsilon$.

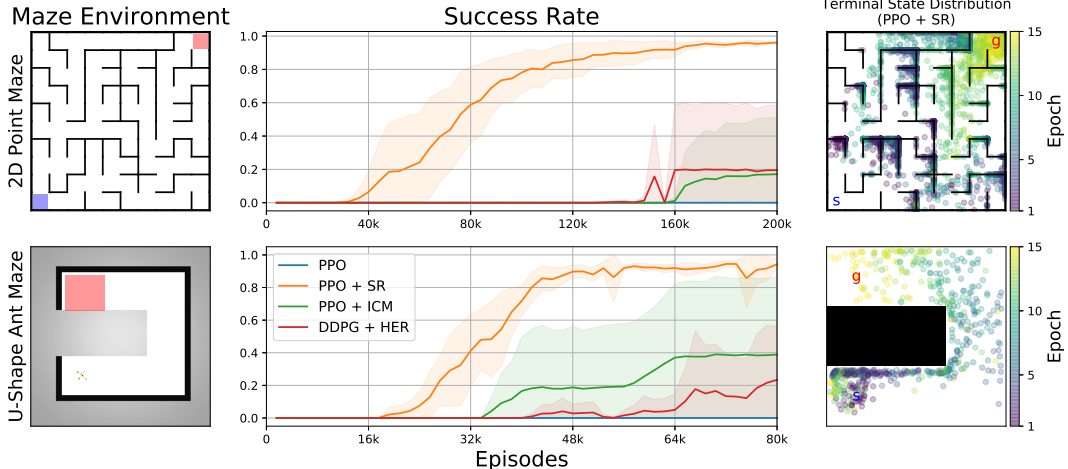

Figure 3: **(Left) Maze environments.** Top row illustrates our 2D point maze; bottom row shows the U-shaped Ant Maze in a Mujoco environment. For the 2D maze, start location is sampled within the blue square; in the ant maze, the agent starts near its pictured location. For both, the goal is randomly sampled from within the red square region. **(Middle) Learning progress.** Lines show rate of goal completion averaged over 5 experiments (shaded area shows mean±SD, clipped to [0, 1]). Only our method (PPO+SR) allows the agent to discover the goal in all experiments. Conversely, PPO with the naive distance-to-goal reward never succeeds. Methods to learn from sparse rewards (PPO+ICM and DDPG+HER) only rarely discover the goals. Episodes have a maximum duration of 50 and 500 environment steps for the 2D Point Maze and Ant Maze, respectively. **(Right) State distribution.** Colored points illustrate terminal states achieved by the policy after each of the first 15 evaluation checkpoints. PPO+SR allows the agent to discover increasingly good optima without becoming stuck in them.

The toy example above (Figure 2) shows an instance of using SR where the base algorithm is A2C, the environment only yields end-of-episode reward ($\gamma = 1$), and the closer rollout $\boldsymbol{\tau}^c$ is only used in gradient estimation when that rollout reaches the goal ($\epsilon = 0$). In our below experiments we mostly use end-of-episode rewards, although SR does not place any restriction on this choice. It should be noted, however, that our method does require that full-episode rollouts are sampled in between parameter updates (based on the choice of treating the *terminal* state of the sibling rollout as $\bar{g}$) and that experimental control over episode conditions ($s_0$ and $g$) is available.[3] Lastly, we point out that we include the state $s_t$, episode goal $g$, and anti-goal $\bar{g}$ as inputs to the critic network $V$; the policy $\pi$ sees only $s_t$ and $g$.

In the appendix, we present a more formal motivation of the technique (Section A), additional clarifying examples addressing the behavior of SR at different degrees of local optimum severity (Section B), and an empirical demonstration (Section C) showing how $\epsilon$ can be used to tune the system towards exploration ($\downarrow \epsilon$) or exploitation ($\uparrow \epsilon$).

# 4   Experiments

To demonstrate the effectiveness of our method, we apply it to a variety of goal-reaching tasks. We focus on settings where local optima interfere with learning from naive distance-to-goal shaped rewards. We compare this baseline to results using our approach as well as to results using curiosity and reward-relabeling in order to learn from sparse rewards. The appendix (Section F) provides detailed descriptions of the environments, tasks, and implementation choices.

**2D Point-Maze Navigation.**   How do different training methods handle the exploration challenge that arises in the presence of numerous local optima? To answer this, we train an agent to navigate a fully-continuous 2D point-maze with the configuration illustrated in Figure 3 (top left). At each point

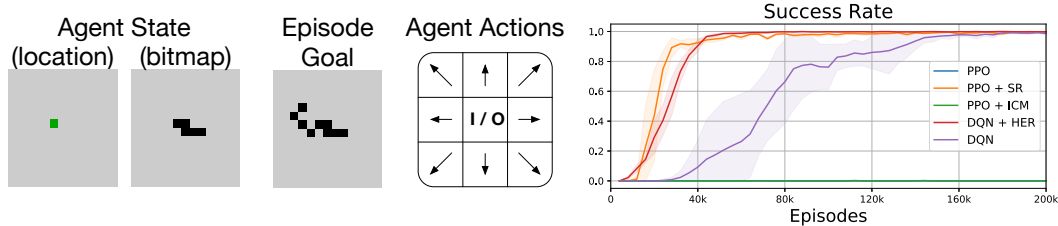

Figure 4: **2D discrete pixel-grid environment.** The agent begins in a random location on a 13x13 grid with all pixels off and must move and toggle pixels to produce the goal bitmap. The agent sees its current location (1-hot), the current bitmap, and the goal bitmap. The agent succeeds when the bitmap exactly matches the goal (0-distance). Lines show rate of goal completion averaged over 5 experiments (shaded area shows mean±SD, clipped to [0, 1]). Episodes have a maximum duration of 50 environment steps.

in time, the agent only receives its current coordinates and the goal coordinates. It outputs an action that controls its change in location; the actual change is affected by collisions with walls. When training using Proximal Policy Optimization (Schulman et al., 2017) and a shaped distance-to-goal reward, the agent consistently learns to exploit the corridor at the top of the maze but never reaches the goal. Through incorporating Sibling Rivalry (PPO + SR), the agent avoids this optimum (and all others) and discovers the path to the goal location, solving the maze.

We also examine the behavior of algorithms designed to enable learning from sparse rewards without reward shaping. Hindsight Experience Replay (HER) applies off-policy learning to relabel trajectories based on achieved goals (Andrychowicz et al., 2017). In this setting, HER [using a DDPG backbone (Lillicrap et al., 2016)] only learns to reach the goal on 1 of the 5 experimental runs, suggesting a failure in exploration since the achieved goals do not generalize to the task goals. Curiosity-based intrinsic reward (ICM), which is shown to maintain a curriculum of exploration (Pathak et al., 2017; Burda et al., 2018a), fails to discover the sparse reward at the same rate. Using random network distillation (Burda et al., 2018b), a related intrinsic motivation method, the agent never finds the goal (not shown for visual clarity). Only the agent that learns with SR is able to consistently and efficiently solve the maze (Figure 3, top middle).

**Ant-Maze Navigation using Hierarchical RL.** SR easily integrates with HRL, which can help to solve more difficult problems such as navigation in a complex control environment (Nachum et al., 2018). We use HRL to solve a U-Maze task with a Mujoco (Todorov et al., 2012) ant agent (Figure 3, bottom left), requiring a higher-level policy to propose subgoals based on the current state and the goal of the episode as well as a low-level policy to control the ant agent towards the given subgoal. For fair comparison, we employ a standardized approach for training the low-level controller from subgoals using PPO but vary the approach for training the high-level controller. For this experiment, we restrict the start and goal locations to the opposite ends of the maze (Figure 3, bottom left).

The results when learning to navigate the ant maze corroborate those in the toy environment: learning from the naive distance-to-goal shaped reward $\tilde{r}$ fails because the wall creates a local optimum that policy gradient is unable to escape (PPO). As with the 2D Point Maze, SR can exploit the optimum without becoming stuck in it (PPO+SR). This is clearly visible in the terminal state patterns over early training (Figure 3, bottom right). We again compare with methods to learn from sparse rewards, namely HER and ICM. As before, ICM stochastically discovers a path to the goal but at a low rate (2 in 5 experiments). In this setting, HER struggles to generalize from its achieved goals to the task goals, perhaps due in part to the difficulties of off-policy HRL (Nachum et al., 2018). 3 of the 5 HER runs eventually discover the goal but do not reach a high level of performance.

**Application to a Discrete Environment.** Distance-based rewards can create local optima in less obvious settings as well. To examine such a setting and to show that our method can apply to environments with discrete action/state spaces, we experiment with learning to manipulate a 2D bitmap to produce a goal configuration. The agent starts in a random location on a 13x13 grid and may move to an adjacent location or toggle the state of its current position (Figure 4, left). We use $L_1$ distance (that is, the sum of bitwise absolute differences). Interestingly, this task does not require the

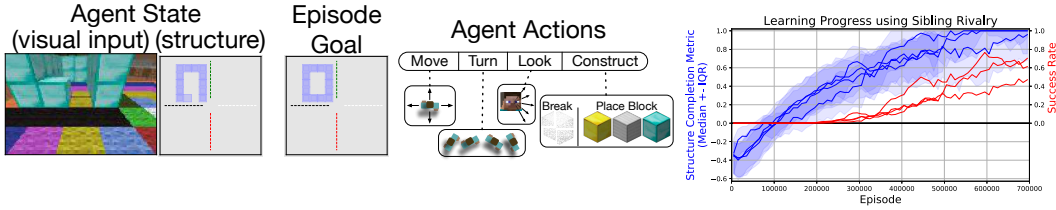

Figure 5: **3D construction task in Minecraft.** The agent must control its location/orientation and break/place blocks in order to produce the goal structure. The agent observes its first-person visual input, the discrete 3D cuboid of the construction arena, and the corresponding cuboid of the goal structure. An episode is counted as a success when the structure exactly matches the goal. The *Structure Completion Metric* is difference between correctly and incorrectly placed blocks divided by the number of goal-structure blocks. In the illustrated example, the agent has nearly constructed the goal, which specifies a height-2 diamond structure near the top left of the construction arena. Goal structures vary in height, dimensions, and material (4806 unique combinations). Episodes have a maximum duration of 100 environment steps.

agent to increase the distance to the goal in order to reach it (as, for example, with the Ant Maze), but naive distance-to-goal reward shaping still creates 'local optima' by introducing pathological learning dynamics: early in training, when behavior is closer to random, toggling a bit from off to on tends to *increase* distance-to-goal and the agent quickly learns to avoid taking the toggle action. Indeed, the agents trained with naive distance-to-goal reward shaping $\tilde{r}$ never make progress (PPO). As shown in Figure 4, we can prevent this outcome and allow the agent to learn the task through incorporating Sibling Rivalry (PPO+SR).

As one might expect, off-policy methods that can accommodate forced exploration may avoid this issue; DQN (Mnih et al., 2015) gradually learns the task (note: this required densifying the reward rather than using only the terminal state). However, exploration alone is not sufficient on a task like this since simply achieving diverse states is unlikely to let the agent discover the task structure relating states, goals, and rewards, as evidenced by the failure of ICM to enable learning in this setting. HER aims to learn this task structure from failed rollouts and, as an off-policy method, handles forced exploration, allowing it to quickly learn this task. Intuitively, using distance as a reward signal automatically exposes the task structure but often at the cost of unwanted local optima. Sibling Rivalry avoids that tradeoff, allowing efficient on-policy learning[4].

**3D Construction in Minecraft.** Finally, to demonstrate that Sibling Rivalry can be applied to learning in complex environments, we apply it to a custom 3D construction task in Minecraft using the Malmo platform (Johnson et al., 2016). Owing to practical limitations, we use this setting to illustrate the scalability of SR rather than to provide a detailed comparison with other methods. Similar to the pixel-grid task, here the agent must produce a discrete goal structure by placing and removing blocks (Figure 5). However, this task introduces the challenge of a first-person 3D environment, combining continuous and discrete inputs, and application of aggressively asynchronous training with distributed environments [making use of the IMPALA framework (Espeholt et al., 2018)]. Since success requires exact-match between the goal and constructed cuboids, we use the number of block-wise differences as our distance metric. Using this distance metric as a naive shaped reward causes the agent to avoid ever placing blocks within roughly 1000 episodes (not shown for visual clarity). Simply by incorporating Sibling Rivalry the agent avoids this local optimum and learns to achieve a high degree of construction accuracy and rate of exact-match success (Figure 5, right).

## 5   Related Work

**Intrinsic Motivation.** Generally speaking, the difficulty in learning from sparse rewards comes from the fact that they tend to provide prohibitively rare signal to a randomly initialized agent. Intrinsic motivation describes a form of task-agnostic reward shaping that encourages exploration by rewarding novel states. Count-based methods track how often each state is visited to reward

reaching relatively unseen states (Bellemare et al., 2016; Tang et al., 2017). Curiosity-driven methods encourage actions that surprise a separate model of the network dynamics (Pathak et al., 2017; Burda et al., 2018a; Zhao & Tresp, 2018). Burda et al. (2018b) introduce a similar technique using distillation of a random network. In addition to being more likely to discover sparse reward, policies that produce diverse coverage of states provide a strong initialization for downstream tasks (Haarnoja et al., 2017; Eysenbach et al., 2019). Intrinsic motivation requires that the statistics of the agent's experience be directly tracked or captured in the training progress of some external module. In contrast, we use the policy itself to estimate and encourage exploratory behavior.

**Curriculum Learning and Self-Play.**  Concepts from curriculum learning (Bengio et al., 2009) have been applied to facilitate learning goal-directed tasks (Molchanov et al., 2018; Nair et al., 2018). Florensa et al. (2018), for example, introduce a generative adversarial network approach for automatic generation of a goal curriculum. On competitive tasks, such as 2-player games, self-play has enabled remarkable success (Silver et al., 2018). Game dynamics yield balanced reward and force agents to avoid over-committing to suboptimal strategies, providing both a natural curriculum and incentive for exploration. Similar benefits have been gained through asymmetric self-play with goal-directed tasks (Sukhbaatar et al., 2018a,b). Our approach shares some inspiration with this line of work but combines the asymmetric objectives into a single reward function.

**Learning via Generalization.**  Hindsight Experience Replay (Andrychowicz et al., 2017) combines reward relabeling and off-policy methods to allow learning from sparse reward even on failed rollouts, leveraging the generalization ability of neural networks as universal value approximators (Schaul et al., 2015). Asymmetric competition has been used to improve this method, presumably by inducing an automatic exploration curriculum that helps relieve the generalization burden (Liu et al., 2019).

**Latent Reward Shaping.**  A separate approach within reward shaping involves using latent representations of goals and states. Ghosh et al. (2018) estimate distance between two states based on the actions a pre-trained policy would take to reach them. Nair et al. (2018) introduce a method for unsupervised learning of goal spaces that allows practicing reaching imagined goal states by computing distance in latent space [see also Péré et al. (2018)]. Warde-Farley et al. (2019) use discriminitive training to learn to estimate similarity to a goal state from raw observations. Learned models have also been applied to perform reward shaping to overcome challenges related to delayed rewards (Arjona-Medina et al., 2019).

## 6   Conclusion

We introduce Sibling Rivalry, a simple and effective method for learning goal-reaching tasks from a generic class of distance-based shaped rewards. Sibling Rivalry makes use of sibling rollouts and self-balancing rewards to prevent the learning dynamics from stabilizing around local optima. By leveraging the distance metric used to define the underlying sparse reward, our technique enables robust learning from shaped rewards without relying on carefully-designed, problem-specific reward functions. We demonstrate the applicability of our method across a variety of goal-reaching tasks where naive distance-to-goal reward shaping consistently fails and techniques to learn from sparse rewards struggle to explore properly and/or generalize from failed rollouts. Our experiments show that Sibling Rivalry can be readily applied to both continuous and discrete domains, incorporated into hierarchical RL, and scaled to demanding environments.

## Footnotes

[1] A straightforward metric, such as $L_1$ or $L_2$ distance, is often sufficient to express goal completion.

[2]Reference implementation available at `https://github.com/salesforce/sibling-rivalry`

[3]Though we observe SR to work when $s_0$ is allowed to differ between sibling rollouts (appendix, Sec. D)

[4]We find that including both sibling trajectories ($\epsilon = \mathtt{Inf}$) works best in the discrete-distance settings

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
