[Supplementary Material]

# A  Algorithm Details

---

**Algorithm 1:** Sibling Rivalry

**Given**

- Environment, Goal-reaching task w/ $S, G, A, \rho(s_0, g), d(,), \delta$ and max episode length
- Policy $\pi : S \times G \times A \to [0, 1]$ and Critic $V : S \times G \times G \to \mathbb{R}$ with parameters $\theta$ and on-policy learning algorithm $\mathbb{A}$
- Inclusion threshold $\epsilon$

**for** *cycle = 1...K* **do**

    Initialize transition buffer $D$

    **for** *episode = 1...M* **do**

        Sample $s_0, g \sim \rho$

        $\boldsymbol{\tau}^a \leftarrow \pi_\theta(...)|_{s_0,g}$      # Collect rollout

        $\boldsymbol{\tau}^b \leftarrow \pi_\theta(...)|_{s_0,g}$      # Collect sibling rollout

        Relabel $\boldsymbol{\tau}^a$ reward using $r'$ and $\bar{s} \leftarrow s_T^b$

        Relabel $\boldsymbol{\tau}^b$ reward using $r'$ and $\bar{s} \leftarrow s_T^a$

        **if** $d(s_T^a, g) < d(s_T^b, g)$ **then**

            $\boldsymbol{\tau}^c \leftarrow \boldsymbol{\tau}^a$

            $\boldsymbol{\tau}^f \leftarrow \boldsymbol{\tau}^b$

        **else**

            $\boldsymbol{\tau}^c \leftarrow \boldsymbol{\tau}^b$

            $\boldsymbol{\tau}^f \leftarrow \boldsymbol{\tau}^a$

        **if** $d(s_T^c, s_T^f) < \epsilon$ ** or ** $d(s_T^c, g) < \delta$ **then**

            Add $\boldsymbol{\tau}^f$ and $\boldsymbol{\tau}^c$ to buffer $D$

        **else**

            Add $\boldsymbol{\tau}^f$ to buffer $D$

    Apply on-policy algorithm $\mathbb{A}$ to update $\theta$ using examples in $D$

---

The above algorithm describes the procedure for integrating Sibling Rivalry into an existing on-policy learning algorithm. We intend for this description to be as general as possible. It should be noted, however, that one requirement of our method is that updates are scheduled according to complete rollouts. This may pose some difficulty for tasks that would benefit from mid-episode updates based on a *transition* horizon rather than the *episode* horizon we implement. This requirement is based on the choice to select anti-goals using the terminal state of the sibling rollout. In future work, we intend to explore the flexibility of this choice and the possibility of using truncated trajectories within our proposed approach.

# B  Controlling the Inclusion Hyperparameter

Sibling Rivalry makes use of a single hyperparameter $\epsilon$ to set the distance threshold for when to include the closer-to-goal trajectory $\boldsymbol{\tau}^c$ in the parameter updates. When $\epsilon = 0$, $\boldsymbol{\tau}^c$ is only included if it reaches the goal. Conversely, when $\epsilon = \texttt{Inf}$, the algorithm always uses both trajectories (while still encouraging diversity through the augmented reward $r'$). We find that this parameter can be used to tune learning towards exploration or exploitation (of the distance-to-goal reward).

This is most evident in the impact of $\epsilon$ on learning progress in the 2D point maze environment, where local optima are numerous (and, in our observation, learning progress is most sensitive to $\epsilon$). For the sake of demonstration, we performed a set of experiments for each of $\epsilon \in [0, 1, ...10]$ distance units. The 2D point maze itself is 10x10, giving us good coverage of options one might consider for $\epsilon$ in this environment. Interestingly, we observe three modes of the algorithm: over-exploration ($\epsilon$ too low), successful learning, and under-exploration ($\epsilon$ too high). We observe these modes to be clearly identifiable using the metrics reported below (Figure 6). In practice a much coarser search over this hyperparameter should be sufficient to identify the optimal range.

Figure 6: **Effect of Inclusion Threshold $\epsilon$ on Sibling Rivalry.** We re-run the 2D point maze experiments using SR with each of the $\epsilon$ settings shown. Rows report success rate, distance to goal, and distance to anti-goal (that is, distance between sibling rollouts) across training for each of the settings. Line plots and heatmap plots provide different views of the same data. This analysis identifies roughly 3 modes of behavior exhibited by our method in this environment. The first, over-exploration, occurs for the lower range of $\epsilon$, where closer-to-goal trajectories are more aggressively discarded. Close inspection shows slow progress towards the goal and a tendency to increase inter-sibling distance (the latter trend appears to reverse near the end of the training window). The second mode corresponds to sucessful behavior: the agent can exploit the distance-to-goal signal but maintains enough diversity in its state distribution to avoid commitment to local optima. The third mode, under-exploration, occurs for the higher range of $\epsilon$, where inclusion of the closer-to-goal trajectory is more permissive. These settings lead the agent to the same pitfall that prevents learning from naive distance-to-goal shaped rewards. That is, it quickly identifies a low-distance local optimum (consistently, the top corridor of the maze) and does not sufficiently explore in order to find a higher-reward region of the maze.

## C   Implementation Details and Experimental Hyperparameters

Here, we provide a more detailed description of the environments, tasks, and training implementations used in our experiments (Section 4). We first provide a general description of the training algorithms as they pertain to our experiments. We follow with task-specific details for each of the environments.

For all experiments, we distribute rollout collection over 20 parallel threads. Quantities regarding rollouts, epochs, and minibatches are all reported *per worker*.

Proximal Policy Optimization (PPO). Many of the experiments we perform use PPO as the backbone learning algorithm. We focus on PPO because of its strong performance and because it is well suited for the constraints imposed by the application of Sibling Rivalry. Specifically, our method requires the collection of multiple full rollouts in between network updates. PPO handles this well as it is able to make multiple updates from a large batch of transitions. While experimental variants that do not use SR do not require scheduling updates according to full rollouts, we do so for ease of comparison. The general approach we employ cycles between collection of full trajectories and multiple optimization epochs over minibatches of transitions within those trajectories. We apply a constant number of optimization epochs and updates per epoch, varying the sizes of the minibatches as needed based on the variable length of trajectories (due to either episode termination after goal-reaching or trajectory exclusion when using SR). We confirmed that this modification of the original algorithm did not meaningfully affect learning.

We standardize our PPO approach as much as possible to avoid results due to edge-case hyperparameter configurations, using manual search to identify such generally useful parameter settings. In the ant maze task, this standardized approach applies specifically to training the high-level policy. We also use PPO to train the low-level policy but adopt a more specific approach for that based on its unique role in our experiments (described below).

For PPO variants, the output head of the policy network specifies the $\alpha \in \mathbb{R}^2$ and $\beta \in \mathbb{R}^2$ control parameters of a Beta distribution to allow sampling actions within a truncated range (Chou et al., 2017). We shift and scale the sampled values to correspond to the task action range. We also include entropy regularization to prevent the policy from becoming overly deterministic early during training.

Table 1: Implementation details for experiments using PPO

| Hyperparameter | Point maze | | | Ant maze (high) | | | Bit flipping | | |
|---|---|---|---|---|---|---|---|---|---|
| | PPO | +SR | +ICM | PPO | +SR | +ICM | PPO | +SR | +ICM |
| Rollouts per Update | 4 | | | 4 | | | 4 | | |
| Epochs per Update | 4 | | | 2 | | | 4 | | |
| m.Batches per Epoch | 4 | | | 4 | | | 4 | | |
| Learning Rate (LR) | 0.001 | | | 0.001 | | | 0.001 | | |
| LR Decay | 0.999 | | | 1.0 | | | 0.999 | | |
| Entropy Reg $\lambda$ | 0.025 | | | 0.025 | | | 0.025 | 0.0 | 0.025 |
| GAE $\lambda$ | 0.98 | | | 0.98 | | | 0.98 | | |
| Bootstrap Value | N | | Y | N | | Y | N | | Y |
| Discount Factor | 1.0 | | 0.98 | 1.0 | | 0.98 | 1.0 | | 0.98 |
| Inclusion thresh. ($\epsilon$) | | 5.0 | | | 10.0 | | | 0.0 | |

Intrinsic Curiosity Module (ICM). We base our implementation of ICM off the guidelines provided in Burda et al. (2018a). We weigh the curiosity-driven intrinsic reward by 0.01 compared to the sparse reward. Note that in the settings we used, ICM is only accompanied by sparse extrinsic rewards, meaning that it only experiences the intrinsic rewards until it (possibly) discovers the goal region. During optimization, we train the curiosity network modules (whose architectures follow similar designs to the policy and value for the given task) at a rate of 0.05 compared to the policy and value network modules.

2D point maze navigation. The 2D point maze is implemented in a 10x10 environment (arbitrary units) consisting of an array of pseudo-randomly connected 1x1 squares. The construction of the maze ensures that all squares are connected to one another by exactly one path. This is a continuous environment. The agent sees as input its 2D coordinates and well as the 2D goal coordinates, which are always somewhere near the top right corner of the maze. The agent takes an action in a 2D space that controls the direction and magnitude of the step it takes, with the outcome of that step potentially

Table 2: Implementation details for off-policy experiments

| Hyperparameter | Point maze | Ant maze (high) | Bit flipping |
|---|---|---|---|
| Rollouts per Update | 4 | | |
| m.Batches per Update | 40 | | |
| m.Batches size | 64 | 128 | 128 |
| Learning Rate (LR) | 0.001 | | |
| Action $L_2$ $\lambda$ | 0.25 | 0.0002 | NA |
| Behavior action noise | $0.1 \times$ action range | | NA |
| Behavior action epsilon | 0.2 | | |
| Polyak coefficient | 0.95 | | |
| Bootstrap Value | Y | | |
| Discount Factor | 0.98 | | |

Table 3: Environment details

| Setting | $S \in$ | $G \in$ | $A \in$ |
|---|---|---|---|
| Point maze | $\mathbb{R}^2$ | $\mathbb{R}^2$ | $[-0.95, 0.95]^2$ |
| Ant maze (high) | $\mathbb{R}^{30}$ | $\mathbb{R}^2$ | $[-5, 5]^2$ |
| Ant maze (low) | $\mathbb{R}^{30}$ | $\mathbb{R}^2$ | $[-30, 30]^8$ |
| Bit flipping | $\{0, 1\}^{13 \times 13 \times 2}$ | $\{0, 1\}^{13 \times 13}$ | $\{0...9\}$ |
| Minecraft | $s^v \in \mathbb{R}^{80 \times 120 \times 3}$, $s^c \in \{0...N_b\}^{11 \times 11 \times 3}$ | $\{0...N_b\}^{11 \times 11 \times 3}$ | $\{0...20\}$ |

Table 4: Task details

| Setting | $m(s)$ | $d(,)$ | $\delta$ | Max. $T$ |
|---|---|---|---|---|
| Point maze | $\mathbb{I}$ | $L_2$ | 0.15 | 50 |
| Ant maze (high) | $[s^0, s^1]$ | $L_2$ | 1.0 | 25 (=500 env steps) |
| Ant maze (low) | $[s^0, s^1]$ | $L_2$ | NA | 20 (env steps) |
| Bit flipping | $s^{:,:,0}$ | $L_1$ | 0.0 | 50 |
| Minecraft | $s^c$ | $\sum x_{ijk} \neq y_{ijk}$ | 0.0 | 100 |

affected by collisions with walls. The agent does not observe the walls directly, creating a difficult exploration environment. For all experiments, we learn actor and critic networks with 3 hidden layers of size 128 and `ReLU` activation functions.

**Ant maze navigation with hierarchical RL.** The ant maze experiment borrows a similar set up to the point maze but trades complexity of the maze for complexity in the navigation behavior. We use this as a lens to study how the different algorithms handle HRL in this setting. We divide the agent into a high-level and low-level policy, wherein the high-level policy proposes subgoals and the low-level agent is rewarded for reaching those subgoals. For all experiments, we allow the high-level policy to propose a new subgoal $g^L$ every 20 environment timesteps. From the perspective of training the low-level policy, we treat each such 20 steps with a particular subgoal as its own mini-episode. At the end of the full episode, we perform 2 epochs of PPO training to improve the low-level policy, using distance-to-subgoal as the reward.

The limits of the maze are $[-4, 20]$ in both height and width. The agent starts at position $(0, 0)$ and must navigate to goal location $g = (x_g, y_g)$ with coordinates sampled within the range of $x_g \in [-3.5, 3.5]$ and $y_g \in [12.5, 19.5]$. It should be noted that, compared to previous implementations of this environment and task (Nachum et al., 2018), we do not include the full range of the maze in the distribution of task goals. For the agent to ever see the sparse reward, it must navigate from one end of the U-maze to the other and cannot bootstrap this exploration by learning from goals that occur along the way. As one might expect, the learning problem becomes considerably easier when this broad goal distribution is used; we experiment in the more difficult setting since we do not wish to impose the assumption that a task's goal distribution will naturally tile goals from ones that are trivially easy to reach to those that are difficult.

At timestep $t$, the high-level controller outputs a 2-dimensional action $a_t \in [-5, 5]^2$, which is used to compute the subgoal $g_t^L = m(s_t) + a_t$. In other words, the high-level action specifies the relative coordinates the low-level policy should achieve. From the perspective of training the high-level policy, we only consider the timesteps where it takes an action and consider the result produced by the low-level policy as the effect of having taken the high-level action.

In all experiments, both the high- and low-level actor and critic networks use 3 hidden layers of size 128 and `ReLU` activation functions.

**2D bit flipping task.** We extend the bit flipping example used to motivate HER (Andrychowicz et al., 2017) to a 2D environment in which interaction with the bit array depends on location. In this setting, the agent begins at a random position on a 13x13 grid with none of its bit array switched on. Its goal is to reproduce the bit array specified by the goal. To populate these examples, we procedurally generate goal arrays by simulating a simple agent that changes direction every few steps and toggles bits it encounters along the way.

We include this example mostly to illustrate (i) that our method can work in this entirely discrete learning setting and (ii) that naive distance-to-goal based rewards are exceptionally prone to even brittle local optima, such as the ones created when the agent learns to avoid taking the toggle-bit action.

We report the (eventually) successful performance using vanilla DQN but point out that this required modifying the reward delivery for this particular agent. In all previous settings, agents trained on shaped rewards receive that reward only at the end of the episode (and no discounting is used). While it is beyond the scope of this work to decipher this observation, we found that DQN could only learn if the shaped reward was exposed at every time step (using a discounting of $\gamma = 0.98$). The variant that used the reward-at-end scheme never learned.

For all bit flipping experiments, we use 2D convolution to encode the states and goals. We pool the convolution output with `MaxPooling`, apply `LayerNorm`, and finally pass the hidden state through a fully connected layer to get the actor and critic outputs.

**3D construction in Minecraft.** To test our proposed method at a more demanding scale, we implement a custom structure-building task in Minecraft using the Malmo platform. In this task, we place the agent at the center of a "build arena" which is populated in one of several full Minecraft worlds. In this particular setting, the agent has no task-specific incentive to explore the outer world but is free to do so. Our task requires the agent to navigate the arena and control its view and orientation in order to reproduce the structure provided as a goal (similar to a 3D version of the bit flipping example but with richer mechanics and more than one type of block that can be placed). All goals specify a square structure made of a single block type that is either 1 or 2 blocks high with corners at randomly chosen locations in the arena. For each sampled goal, we randomly choose those configuration details and keep the sampled goal provided that it has no more than 34 total blocks (to ensure that the structure can be completed within a 100 timestep episode). The agent begins each episode with the necessary inventory to accomplish the goal. Specifically, the goal structures are always composed of 1 of 3 block types and the agent always begins with 64 blocks of each of those types. It may place other block types if it finds them.

The agent is able to observe the first-person visual input of the character it controls as well as the 3D cuboid of the goal structure and the 3D cuboid of the current build arena. The agent therefore has access to the structure it has accomplished but must also use the visual input to determine the next actions to direct further progress.

The visual input is process through a shallow convolution network. Similarly, the cuboids, which are represented as 3D tensors of block-type indices, are embedded through a learned lookup and processed via 3D convolution. The combined hidden states are used as inputs to the policy network. The value network uses separate weights for 3D convolution (since it also takes the anti-goal cuboid as input) but shares the visual encoder with the policy.

Owing to the computational intensity and long run-time of these experiments, we limit our scope to the demonstration of Sibling Rivalry in this setting. However, we do confirm that, like with the bit flipping example, naive distance-to-goal reward shaping fails almost immediately (the agent learns to never place blocks in the arena within roughly 1000 episodes).

For the work presented here, we compute the reward as the change in the distance produced by placing a single block (and use discounting of $\gamma = 0.99$). We find that this additional densification of the reward signal produces faster training in this complex environment.