[Reviews · NeurIPS 2019]

Reviewer 1



The scores below are consistent with "overall score" legend. Originality: 7. 1) The reviewer found this method novel and interesting. 2) This work clearly differs from the previous work by introducing exploration reward which pushes the policy away from the local optimum of the shaped rewards. 3) The related work is reasonably cited. Quality: 4. The reviewer found the technical contribution not well-analyzed. 1) From the described algorithm, it is not fully clear how it deals with the case of a long narrow corridor with the goal at one end and the starting point at the other end. -----------------------------start goal----------------------------- Assume policy ended up in far enough points in sibling rollouts but didn't reach the goal. If the policy is pushed back from the final state of the rollout which came closer to the goal (by learning from tau^f), how does it make progress along the corridor and not gets stuck by returning to the origin? If the only force from making this happen is the original shaped reward, then the test case of two long parallel corridors connected in a U-shape should fail: -----------------------------start| | --------------------------------| | | | --------------------------------| -----------------------------goal| The reviewer would suggest running those two test cases and verifying that the algorithm can solve both. 2) It is not clear what is the purpose of learning from tau^c, which seems to be correlated with the original shaped reward (pushing the "close" rollout end state away from the "far" rollout end state). Would it make more sense to create a balancing coefficient between the original reward and the tau^f part? 3) L161 "we often wish to allow learning from both tau^f and tau^c" -- it is not clear to the reviewer how such learning is supposed to work. If an agent is standing in a corridor, one term would encourage it to go forward and the other -- to go backward. How is learning from both productive? 4) The statement in lines 104-105 seems incorrect: the described shaped reward could have a different global optimum to the original unshaped reward. This potentially makes the shaped reward baseline weak. Imagine we start with distance epsilon to the goal. Let's assume optimal path requires first increasing the distance to d and then decreasing to 0. This could happen if the goal is on the other side of the wall with respect to the starting point (the parallel corridor example above illustrates the point). The return of the policy staying at the starting position is better than the original optimal policy for some values of gamma, epsilon and ranges of d: the agent would be getting very small negative rewards the whole episode instead of getting some very large negative rewards some part of the episode. This problem could potentially be fixed by taking increments as reward instead of absolute distance values. However, this is not discussed in the paper. 5) One of the relevant and very simple baselines is missing. One of the recent papers, "Episodic Curiosity through Reachability" (ICLR'19), introduced a Grid Oracle baseline. Essentially, an agent is given reward for visiting xy-grid cells. How would it work in the navigation environments used for this paper? 6) It is not clear if the paper conducted experiments with unshaped sparse rewards. Clarity: 5. The reviewer found the algorithm description not easy to understand. Major points: 1) "Solving Sparse Reward Tasks Using ..." in the paper name is too general. In reality, the paper discusses only goal-conditioned problems. 2) It is hard to understand from the main text which reward function from (9) is always used for gradient calculation and which not. The supplementary material helps, but the reviewer would suggest making it clear in the main text. 3) L138: "We hypothesize ..." -- is there any evidence supporting the claim? For example, a theoretical or experimental proof. 4) L88: something is missing at the beginning of the line. 5) L90-92: it would be better to remove the note and apply m-function such that the text above is mathematically correct. Minor points: 1) L12: "curiosity ... exhibit poor performance" -- this is a very general statement in the abstract, however, the paper compares only to two methods relying on prediction error: ICM and RND. It would be worth mentioning those methods specifically without generalizing to all curiosity methods. 2) L36: problem-specific Significance: 5. The reviewer would suggest doing additional experiments to better analyze the strong/weak sides of the algorithm -- which are currently not fully clear. 1) The method seems limited to the environments where it is possible to obtain siblings trajectories. However, in many environments it might not be possible -- which limits the generality of the method. As one example, DMLab starts every new episode with its own map. Is there any generalization possible to address this? If not, this limitation should be stated more clearly throughout the paper -- along with some arguments on why such environments are interesting/important. 2) Although the paper uses 4 different domains for experiments, the experiments in every domain are not diverse or revealing for the properties of the algorithm. In particular, the reviewer would suggest doing the experiments with one long corridor and two connected long corridors described above. -------------------------------------------------------------------------- -------------------------------------------------------------------------- POST-REBUTTAL UPDATE: Overall, I'm going to increase my score from 5 to 6 because the authors put in some effort to address my concerns. If the paper is accepted, I would strongly suggest addressing the following points in the camera-ready version: 1) It is very important for the quality of the paper to add the Grid Oracle baseline. It is easy to implement and there is no obvious reason to omit it. Since the paper uses xy-coordinates, rewarding visiting xy coordinate grid cells is the most natural thing to do. Without this baseline, it is hard to fully judge the value of the method. 2) U-shaped experiment added by the authors alleviates some of my concerns. Nevertheless, I still don't fully understand why the method works in this case. It would be great if there is an extended discussion of this question in the camera-ready version of the paper (supplementary material is also fine). 3) The setting where all episodes start at exactly the same position is quite artificial. Additional evidence is needed to establish that noise in the starting conditions doesn't break the algorithm. It would be great to have such an experiment.

Reviewer 2



This paper address the problem of sparse reward learning where a biased but helpful distance-to-goal metric is available at every state. The difficulty in solving these problems that directly optimizing for minimum distance to goal can often lead to poor local minima, but only relying on the sparse reward is too data inefficient for practical use. Originality: The paper proposes using "sibling" trajectories to avoid falling into local minima. Sibling trajectories are trajectories that share the same goal, start state, and policy. Instead of optimizing the distance to goal, each trajectory is relabled with an "anti-goal", taken from the sibling trajectory, and the reward is to minizme distance to goal while maximizing distance to anti-goal. This method would increase the diversity of states visited in order to find optimal paths. However, this new reward function is a heuristic and is not guaranteed to return the optimal policy with respect to the original sparse reward. The method is novel to my knowledge. Quality: The method performs very well in the maze and ant experiments, which compare to HER and ICM. I would be curious to see how a maximum entropy RL method would perform on these environments (SAC/SQL). The Minecraft environment performance is not compared to any other method, this should be corrected. The SR method does not seem to hurt final performance compared to other methods. Unlike most Goal-conditioned RL problem statements, this method does aim to solve the any-state-to-any-state problem, and does not compare to these. Clarity: The "Self-balancing reward" section is not clear. The purpose of the derivation is unclear given that the authors can only hypothesize that "the original gradient definition [..] obeys a similar relationship" with what they are optimizing. Other than that, the paper is readable. A better explanation of how this method avoids local minima that discusses the difference between V(s,g) and d(s,g). Does this method do anything more than simply increase the diversity of states? Significance: The problem of only having heuristic distance functions that do not directly correlate to the state value is a significant one. It would be useful to experiment with how bad a distance can be where this method still works. Overall, I find the method interesting. However, the method assumes that use can choose what initial state to reset the environment to (in order to generate sibling trajectories from the same s_0). This property is not generally assumed for MDPs, and may lead to this method being difficult to use in the real world, or in environments that have much wider initial state distributions. It would be good for the authors to discuss this assumption. ----------- After the rebuttal and discussion, I have decided to increase my score to 7. While the resetting is a big assumption, other papers have made this kind of assumption and it is feasible for a number of settings. The experiments cover a large variety of types of environments. I could personally compare to the method in my future work. I think this work is interesting, as long as the authors clarify the method section.

Reviewer 3



This paper proposed Sibling Rivalry, a simple yet efficient method for learning goal-reaching tasks from distance-based shaped rewards. Sibling Rivalry introduces a pair of rollouts to encourage exploration while destabilize local optima, thus it is able to tackle sparse reward problems efficiently. The extensive experiments demonstrate Sibling Rivalry's success on a variety of sparse reward problems. The idea is very interesting and simple. Self-balancing shaped reward (Eq.3) strikes a balance between exploiting available rewards and exploring diverse states. Sibling Rivalry samples pair of rollouts and introduces mutual relabeling based on the self-balancing rewards, decides to select which rollouts for policy gradient estimation. Furthermore, the inclusion hyper-parameter $\epsilon$ is able to control the over-exploration and under-exploration. However, we need to search for this parameter, while can not learn it adaptively. The experiments are carried out on a variety of continuous and discrete sparse reward tasks such as maze navigation, 3D construction in Minecraft and so on. It is clearly that Sibling Rivalry incorporating with on-policy learning method PPO is able to achieve the best successful rate compared to DDPG+HER and other baselines. However, it is better to provide more results such as the cumulative reward curves w.r.t steps of interactions to learn the sample complexity. The paper is well organized and clearly written. The Sibling Rivalry provides an efficient method to tackle sparse reward problems and will attract the attentions from sparse reward research community. I have read the rebuttal which addressed my concern, and other reviews.

[Author Response · NeurIPS 2019]

We thank the reviewers for their thoughtful feedback, and note the apparent consensus that our contribution, Sibling
Rivalry (SR), is interesting and novel. We would like to emphasize that, as several reviewers have observed, SR
is **simple to implement and apply**, **works across a variety of settings, and learns from generic distance-based**
**shaped rewards that do not rely on domain-expertise.** To make SR a valuable resource for the research community,
**we will release an open-source implementation that shows SR is light-weight** ($\sim$ 50 lines of added code).

**Revisions**    We will fix all writing issues and revise the title and notation. We will clarify how trajectories are selected
for gradient computation and that results with only sparse rewards are omitted because they all fail in our setting.

**Intuition (R1, R2)**    Reward shaping presents significant challenges for on-policy learning of goal-conditioned policies,
as such methods often converge to sub-optimal policies [Williams 1992]. Such "local optima" are, e.g., low-entropy
policies that fail to reach goal-states and get stuck in regions away from the goal. Manual reward shaping can prevent
such outcomes, but is often domain-specific, ad-hoc, and challenging. Instead, SR uses pairs of rollouts to automatically
estimate when policies get stuck in local optima and prevents the policy's terminal-state distribution $\rho_g^\pi(s_T)$ from
collapsing around the associated regions of state space. In effect, this gives a **dynamic way to encourage exploration**
**away from local optima** while continuing to guide $\rho_g^\pi(s_T)$ towards the goal $g$. In addition, the SR reward function
converges to the sparse reward as the policy learns to reach the task-goal states, preserving the underlying task definition.

**Assumptions**    The key requirements for applying SR are (1) that goal completion can be expressed via a distance
metric and (2) that the distance metric and control over episode start state/goal are available during learning. In
simulated environments, this availability should exist. We will clarify these requirements and how they are met in
our experiments. In real-world environments, it may be impossible to sample sibling rollouts according to their exact
definition; we consider sensitivity to noise in the sampling conditions as a direction for future research.

**We will include the above and clarify the existing exposition in the paper, with particular attention to improving**
**the presented motivation.** Similarly, we will discuss challenges more clearly, such as sensitivity to $\epsilon$ in the U-maze
(below).

**Additional Experiments (R1, R2)**    We have performed the suggested Corridor and U-shaped maze experiments: the
results strengthen the intuition above and reaffirm the effectiveness of SR. We will add the results to the Appendix. **SR**
**solves the Corridor task easily at all lengths tested** (from length 5 to 25) and handles the U-maze task well for the
ranges tested (total maze length from 7 to 31). In comparison, **both ICM and HER fail to solve the longer versions**
**of the tasks.** In the most extreme U-maze case, the local optimum is on average $\sim 1.5$ distance units from the goal but
successful navigation requires first moving roughly $10\times$ that distance away from the goal. In this case, the range of
inclusion hyperparameter ($\epsilon$) values that produce good results with SR are limited ($0.2 \leq \epsilon \leq 0.8$). Due to the limited
rebuttal period, we are unable to report all suggested baseline comparisons here but will include them in the paper.

**R1**    *...not fully clear...long narrow corridor...U-shape should fail...* Thank you for the insightful suggestion, we hope
the above results clarify the robustness of SR. We observed that the two distance terms in the SR reward do not cancel
out (see the Corridor results); rather, it is more helpful to think of their distinct effects on the distribution of terminal
states $\rho_g^\pi(s_T)$. The distance-to-goal reward draws $\rho_g^\pi(s_T)$ towards $g$ while the distance-to-sibling reward prevents the
former reward from collapsing $\rho_g^\pi(s_T)$ around hurtful optima.

*...different global optimum...shaped reward baseline weak...* Because we only supply shaped rewards at the terminal
timestep, the concern about the naive shaped rewards being a weak baseline does not apply. Nevertheless, we will
clarify that providing the absolute distance values as reward at each timestep may distort the global optimum.

**R2**    *...the difference between $V(s, g)$ and $d(s, g)$...how bad a distance can be...* The experiments described above
clarify how the quality of the distance function impacts SR. Note that local optima hinder learning even with "good"
distance functions, as shown in our bit-flipping and Minecraft experiments.

*...any-state-to-any-state problem...* We avoid the any-state-to-any-state version of the goal-conditioned RL problem in
order to address the realistic scenario where task-relevant goals only occupy a small portion of the state space.

**R3**    *Sample complexity* We will plot results in terms of sampled timesteps rather than episodes. The number of episodes
provides an upper bound on the number of timesteps (`#timesteps` $\leq$ `#episodes` * `MaxEpisodeLength`). Hence,
plotting results in terms of timesteps will shift the learning curves to the left. **We confirmed that SR significantly**
**outperforms baselines also when rewards are plotted vs timesteps.**

*Formal analysis* Our work contributes an extensive empirical validation of SR, we leave formal analysis for future work.

[Meta-Review · NeurIPS 2019]

The strength of the paper lies in proposing a novel method which is interesting, simple, novel, yet effective. On the other hand, the reviewers were concerned that the proposed approach is somewhat heuristic, and it is unclear why it works and what its limitations are. Specifically, the reviewers were unclear why the method helps in the corridor and u-shaped maze experiments; a much more thorough analysis of these cases would be greatly beneficial to the reader. We recommend to add additional baselines (such as a grid-oracle baseline), experiments (such as adding noise to the starting condition), and analysis of results, as suggested by reviewers. The authors should also clarify some points in the "self-balancing reward" section that were unclear to the reviewers.